# Postsynaptic plasticity of Purkinje cells in mice is determined by molecular identity

Stijn Voerman [1,3], Bastiaan H. A. Urbanus[1,3], Martijn Schonewille[1], Joshua J. White [1] & Chris I. De Zeeuw [1,2 ✉]

Cerebellar learning is expressed as upbound or downbound changes in simple spike activity of Purkinje cell subpopulations, but the underlying mechanism remains enigmatic. By visualizing murine Purkinje cells with different molecular identities, we demonstrate that the potential for induction of long-term depression is prominent in downbound and minimal in the upbound subpopulation. These differential propensities depend on the expression profile, but not on the synaptic inputs, of the individual Purkinje cell involved, highlighting the functional relevance of intrinsic properties for memory formation.

[1] Department of Neuroscience, Erasmus MC, 3000 DR Rotterdam, The Netherlands. [2] Netherlands Institute for Neuroscience, 1105 CA Amsterdam, The Netherlands. [3] These authors contributed equally: Stijn Voerman, Bastiaan H. A. Urbanus. ✉email: c.dezeeuw@erasmusmc.nl

Different Purkinje cell (PC) subpopulations in which the simple spike activity is bound to go up or down during cerebellar learning are located in so-called upbound and downbound microzones, respectively[1]. Classical learning paradigms that are controlled by these upbound and downbound microzones include for example adaptation of the vestibulo-ocular reflex[2,3] and Pavlovian eyeblink conditioning[4–6], respectively. Recent advanced applications in single-cell RNA sequencing have revealed that PCs differentially express hundreds of genes that are associated with either aldolase C or PLCB4[7–9], and that many of these genes may contribute to the intrinsic electrophysiological differences between the upbound and downbound microzones[1,10]. However, it remains to be elucidated whether and how differences in long-term depression (LTD), which is arguably the most studied form of cerebellar plasticity[11–13], differentially emerge in upbound and downbound PCs.

## Results and discussion

Here we investigated LTD-induction at the parallel fiber (PF) to PC synapse in sagittal slices of the cerebellar cortex of mice that express GFP linked to excitatory amino-acid transporter 4 (EAAT4)[14]. EAAT4 is highly expressed in Aldolase C-positive, but not PLCB4-positive (PLCB4+), PCs (Fig. 1a, b, Suppl. Data 1-3), allowing us to visualize these subpopulations in real time and investigate the propensity for LTD-induction in relation to the downbound or upbound nature of individual PCs[1]. We aimed to match physiological conditions as much as possible in vitro by using 1) adult mice (>P40) with a fully developed microzonal organization[15], 2) direct climbing fiber (CF) stimulation rather than the more artificial somatic depolarization[16], and 3) an interval of PF to CF stimulation (120 ms) that is compatible with the timing of increased complex spike firing observed in conditioning in intact mammals[5,17–19]. In the presence of picrotoxin to block inhibition and following identification of the level of EAAT4 expression, we induced postsynaptic LTD in 22 PCs, distributed across the lobules of the cerebellar cortex (Fig. 1c-e). Although EPSC amplitude following LTD-induction was significantly ($p < 0.01$; t-test) reduced in EAAT4-positive (EAAT4+) PCs ($n = 11$), it was significantly more reduced in EAAT4-negative (EAAT4-) PCs ($n = 11$) compared to EAAT4+ PCs with the use of the same protocol (EAAT4-: $0.75 \pm 0.03$ | EAAT4+: $0.92 \pm 0.02$, $p = 0.0008$), (Fig. 1f–h, Suppl. Data 4). When comparing PC pairs within the same lobule, we still observe a stronger reduction in EPSC size in EAAT4- PCs (EAAT4-: $0.78 \pm 0.04$ | EAAT4+: $0.94 \pm 0.02$, $p = 0.018$) (Fig. 1h). The paired pulse ratio (PPR) before and after LTD-induction did not differ among EAAT4- and EAAT4+ PCs (Suppl. Figure 1a,b, Suppl. Table 1), providing evidence for a postsynaptic nature of the plasticity induced. Likewise, additional control analyses of series and membrane resistances, holding currents as well as size of the complex spikes did not yield any difference among the cell groups either (Suppl. Figure 1b–e, Suppl. Table 1). Additionally, we find no significant correlation between the size of the complex spike and the amount of LTD induced, in both the EAAT4+ ($p = 0.82$) and EAAT4- ($p = 0.73$) groups (Suppl. Figure 1e, Suppl. Data 4).

Our findings are in line with previous studies on regional differences in LTD-induction[20,21] as well as regional and cellular differences in postsynaptic LTP-induction[22], but it remains to be elucidated to what extent differences in presynaptic input versus intrinsic properties contribute to differences in the downbound or upbound nature of PCs. We therefore aimed to isolate the presynaptic element by stimulating PFs at a high frequency in coronal slices at room temperature, which is known to increase the primary effector of presynaptic LTP, nitrogen monoxide[23] (Fig. 2a, b, d, Suppl. Figure 2, Suppl. Data 5). In contrast to our findings on LTD, we did not find any significant difference between EAAT4+ PCs ($n = 11$) and EAAT4- PCs ($n = 11$) in normalized EPSC amplitude (EAAT4-: $1.13 \pm 0.09$ | EAAT4+: $1.16 \pm 0.08$, $p = 0.80$) (Fig. 2b, c). Here too, the PPR did not differ between EAAT4+ and EAAT4- PCs (Fig. 2c). Next, aiming to replicate our findings under different conditions, we also induced presynaptic LTP in sagittal slices at physiological temperature (Fig. 2d, e, Suppl. Data 6). In none of these control conditions did we find any significant difference between PC subpopulations (Suppl. Table 2), suggesting presynaptic input does not determine the plasticity differences between EAAT4+ and EAAT4- PCs.

Potential input differences such as vesicle release, number of synapses and/or kinetics of postsynaptic receptors might also contribute to differences in synaptic plasticity between PCs[20]. To further explore this possibility, we recorded miniature excitatory postsynaptic currents (mEPSCs, $n = 66$, 40+/26-) and miniature inhibitory postsynaptic currents (mIPSCs, $n = 88$, 46+/42-) from EAAT4+ PCs and EAAT4- PCs (Fig. 3a–d, Suppl. Figure 3). Analyses of miniature events did not reveal any difference in frequency, amplitude, and rise or decay time of neither mEPSCs nor mIPSCs between EAAT4+ PCs and EAAT4- PCs (Fig. 3e, f; Suppl. Table 3, Suppl. Data 7-11). These results support the hypothesis that heterogeneity in PC plasticity, including that of postsynaptic LTD, largely reflects differential intrinsic properties of PC subpopulations[1,7,8].

Our plasticity data highlight that individual PLCB4+ PCs, which are negative for EAAT4, differentially respond to similar inputs. Indeed, when LTD is induced by stimulating the CF input 120 ms after PF stimulation, the expression of LTD in EAAT4- PCs is enhanced compared to that in EAAT4+ PCs. Even when PLCB4+ PCs are surrounded, in a central or posterior lobule, by EAAT4+ PCs and thus putatively receive the same PF input, they still adhere to the cell physiological properties that come with their intrinsic protein expression pattern. Likewise, individual EAAT4+ PCs in the anterior lobule with largely EAAT4-, i.e., PLCB4+, PCs, also adhere to their intrinsic propensities, showing a relatively low capacity for LTD-induction. Despite these compelling data, we must be careful to conclude that this difference is purely related to the intrinsic properties of PCs. As there are many different protocols that can effectively induce LTD at PF-PC synapses, a difference in LTD capacity might not be observed using different intervals of PF-CF stimulations[17,24,25]. Additionally, there is evidence to suggest that multiple PF and CF signals are important or even required for LTD induction, in contrast to our single PF-CF signal[24,26,27].

Our findings have important implications for cerebellar learning, which is in many cases likely to be mediated synergistically by neighboring, downbound and upbound microzones. For example, the downbound microzone in lobule simplex containing PCs that show simple spike suppression during the expression of a conditioned eyeblink response is bordered by an upbound microzone that shows facilitation during the same training trials, possibly encoding different kinematic parameters[1,4]. We demonstrate that PCs do not all follow the same computation on differential inputs, but rather feature intrinsic differences that regulate differential plasticity outcomes to the same input. Future work in awake behaving animals should elucidate to what extent side-by-side subpopulations with distinct molecular identities indeed correspond to upbound and downbound microzones, to what extent each of these subpopulations contributes to cerebellar learning and to what extent differential propensities for postsynaptic LTD- and LTP-induction contribute to this process.

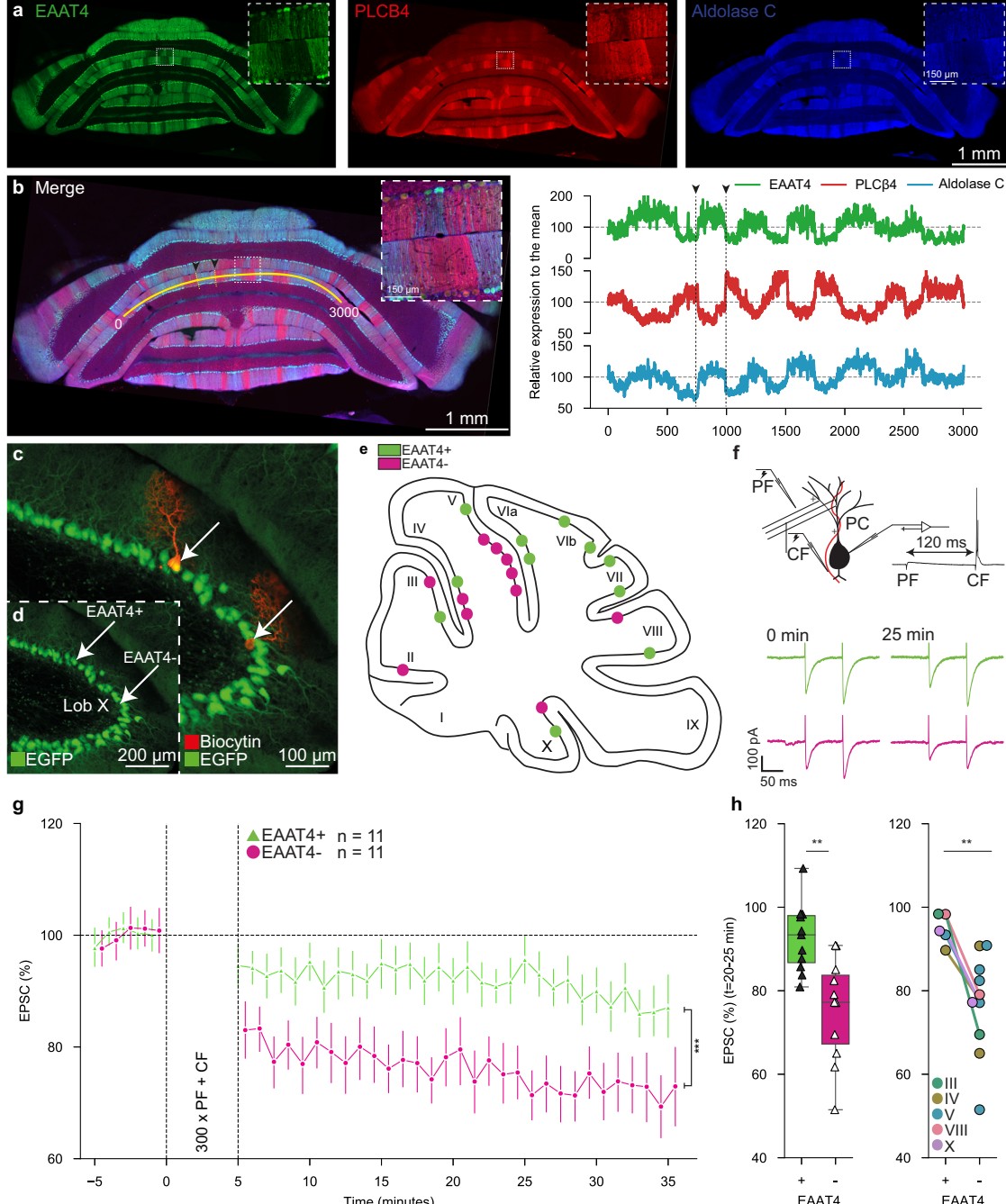

**Fig. 1 EAAT4-/PLCB4+ PCs exhibit stronger LTD than EAAT4+/PLCB4- PCs. a** Coronal cerebellar sections show the expression patterns of EAAT4 (left), PLCB4 (middle) and AldoC (right). Insets, magnification of lobule VII/VIII. EAAT4 expression is complementary to PLCB4 expression and follows AldoC expression. **b** Merge of images in a (left), used in the quantification of expression (right). Normalized expression of EAAT4, PLCB4, and AldoC (right). Black dotted line represents the yellow dotted line in the left panel. **c, d** Example confocal image of a sagittal slice with two patched and labelled PCs (arrows) in lobule X, one EAAT4+ and one EAAT4-. **e** Map of all cells included in LTD analysis, sorted by their EAAT4 expression. In total, 11 EAAT4+ cells and 11 EAAT4- cells are mapped. **f** Schematic representation of the LTD induction protocol (top), and example traces of EPSCs before and 25 minutes after LTD induction (bottom). **g** Time-course of the EPSC amplitude over time. LTD induction occurred between 0 and 5 minutes. LTD induction was significantly stronger in EAAT4- PCs. EPSC amplitude is normalized to the mean value before LTD induction. Error bars represent 95% confidence interval (CI). **h** Normalized EPSC size 25 minutes after LTD induction (left). When comparing only sets of PCs within the same lobule, LTD was still stronger in EAAT4- PCs (right) (EAAT4+ $n = 5$, EAAT4- $n = 10$). Bars and boxes represent a 25th percentile. *** $= p < 0.001$, ** $= p < 0.01$. Independent samples t-test was used for data presented in 1g,h (left). Mann-Whitney U-test was used for data presented in panel 1h (right) (Suppl. Table 1).

## Methods

**Animals**. In the present study, we utilized EAAT4 as a marker for PC subpopulations. The expression of EAAT4 is correlated with that of Aldolase C (Fig. 1; see also ref. [28]). In order to identify EAAT4 expression in vitro, we utilized an EAAT4-eGFP-reporter mouseline. This allowed for the identification of EAAT4+ and EAAT4- PCs in both

sagittal and coronal slices. Adult animals aged >P40 were used in all in vitro electrophysiology experiments (mean age 132 days). Both male and female mice were used. All animal experiments were performed in accordance with the guidelines of the Dutch Ethical Committee for animal experiments, and in accordance with the Animal Welfare Board of the Erasmus MC, in line with Dutch and EU legislation.

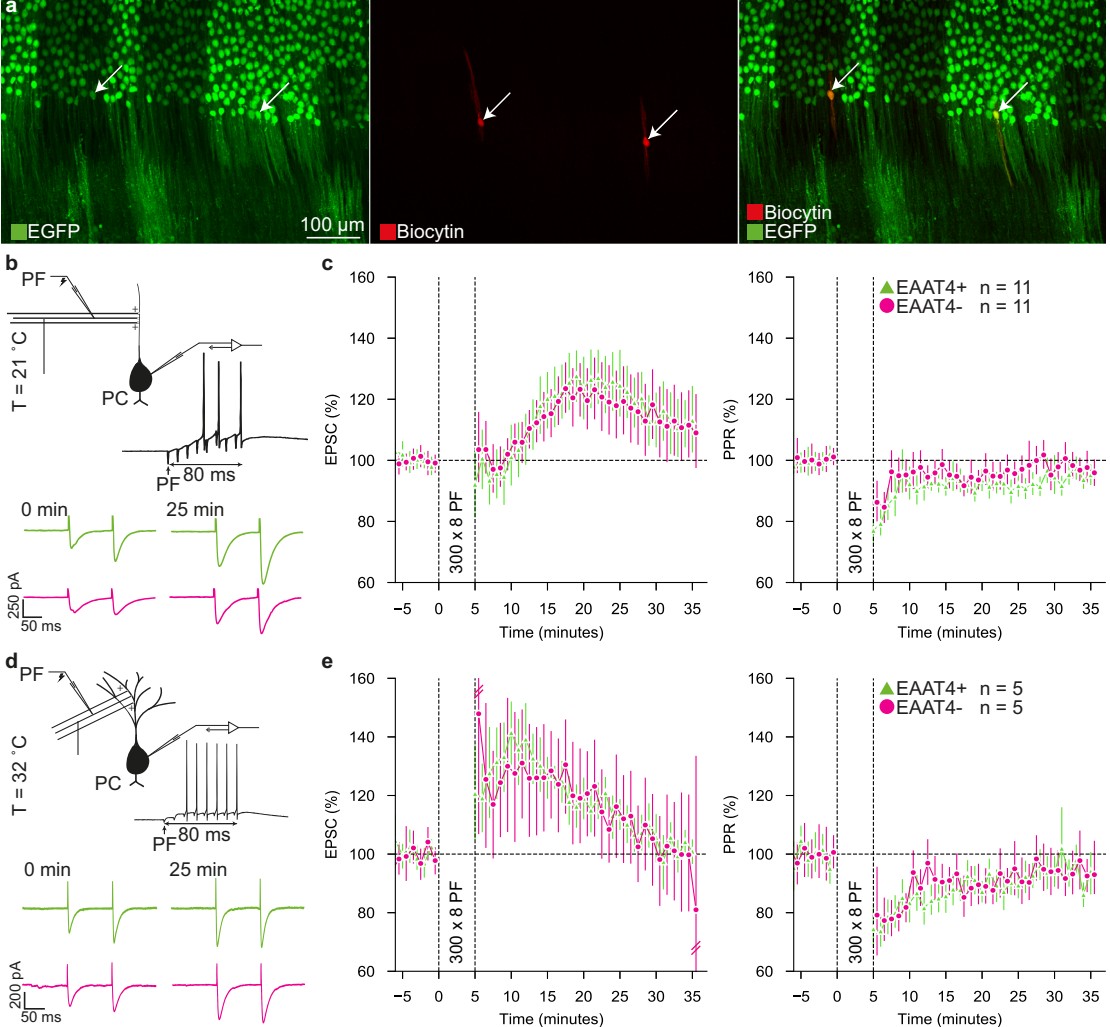

**Fig. 2 Presynaptic LTP is not different between EAAT4+ and EAAT4- PCs. a** Example confocal images of a coronal slice with two patched PCs (middle, right), one PC is EAAT4+ and the other EAAT4-. **b** Schematic representation of LTP induction (top), and example traces of EPSCs before and 25 minutes after LTP induction. **c** Time-course of the EPSC amplitude over time (left). EPSC amplitude was similar after in LTP induction for both EAAT4+ and EAAT4- PCs (both *n* = 11). Additionally, we observed no differences in the PPR (right) over the course of the experiment. Error bars represent 95% CI. Values were normalized to the mean before LTP induction. **d**, **e** Same as in b-c, but in sagittal slices at physiological temperature. At physiological temperature, neither the LTP-induced EPSC amplitude nor PPR differs between EAAT4+ and EAAT4- PCs (both *n* = 5). Independent samples t-test was used to determine differences for all statistical data presented in panels c and e (Suppl. Table 2).

**Tissue preparation**. Mice were decapitated under isoflurane anesthesia, the cerebellum was removed, and subsequently moved into an ice cold 'slicing solution'. The slicing solution contained (in mM) 240 sucrose, 2.5 KCl, 1.25 $Na_2HPO_4$, 2 $MgSO_4$, 1 $CaCl_2$, 26 $NaHCO_3$, and 10 D-Glucose. Either coronal or sagittal (vermis) slices, both 250 $\mu$m thick, were cut using a vibratome (Leica VT1200S). Coronal slices were used in mEPSC, mIPSC, and LTP experiments. Sagittal slices were used in LTD and LTP experiments. Slices were incubated within a carbogenated artificial cerebrospinal fluid (ACSF) containing (in mM) 124 NaCl, 2.5 KCl, 1.25 $Na_2HPO_4$, 2 $MgSO_4$, 2 $CaCl_2$, 26 $NaHCO_3$, and 15 D-Glucose for at least 1 hour at ±34 °C.

**Electrophysiology**. All in vitro electrophysiology was performed with EPC10-USB amplifiers (HEKA electronics, Lambrecht, Germany) using Patchmaster software. Purkinje cells were visualized using upright microscopes (Axioskop 2 FS, Carl Zeiss Microscopy GmbH, Jena, Germany). Additionally, microscopes were optionally equipped with fluorescent LED lights (Colibri 7, Carl Zeiss Microscopy, Göttingen, Germany), in order to visualize eGFP expression in PCs.

Recording electrodes were within the range of 2-5MΩ (OD 1.65 mm, ID 1.11 mm, World Precision Instruments, Sarasota, FL, USA). The recording electrodes were prepared using a P-1000 micropipette puller (Sutter Instruments, Novato, CA, USA). Electrodes were filled with different intracellular solutions depending on the specific experiment. First, for miniature EPSC (mEPSC) recordings we used a cesium methanosulfonate-based internal solution containing

(in mM) 130 $CsMeSO_4$, 4 $MgCl_2$, 0.2 EGTA, 10 HEPES, 4 $Na_2ATP$, 0.4 $Na_3GTP$, 10 phosphocreatine-disodium, 1 QX-314 (pH 7.25–7.35, osmolarity 300 ± 5). Second, for miniature IPSC (mIPSC) recordings we used a cesium chloride based internal solution containing (in mM) 150 CsCl, 1.5 $MgCl_2$, 0.5 EGTA, 4 $Na_2ATP$, 0.4 $Na_3GTP$, 10 HEPES, 5 QX-314 (pH 7.25–7.35, osmolarity 300 ± 5). Third, during LTP and LTD recordings we used a potassium gluconate-based internal solution containing (in mM) 120 K-Gluconate, 9 KCl, 10 KOH, 4 NaCl, 10 HEPES, 28.5 Sucrose, 4 $Na_2ATP$, 0.4 $Na_3GTP$ (pH 7.25–7.35, osmolarity 300 ± 5).

**PF-PC long-term plasticity**. PF-PC plasticity experiments included long-term potentiation (LTP) and long-term depression (LTD). A subset of LTP experiments were performed in coronal slices, and both LTD and a subset of LTP experiments in sagittal slices. Both LTP and LTD were performed with 100 $\mu$M of picrotoxin (PTX, Sigma-Aldrich, Missouri, United States) added to the ACSF. Directly adjacent PCs were avoided, due to the likeliness of inducing plasticity in adjacent PCs, and possibly altering the result of the last cell recorded. Stimulation electrodes were filled with external saline. Throughout LTP and LTD experiments, series resistance was monitored. Recordings were excluded from analysis if series resistance changed by >25% over the course of the experiment, or if their EAAT4 identity could not be confirmed. Additionally, cells were excluded if the EPSC size trended downwards before LTD induction, as this indicates a reduction in EPSC size not associated with LTD. Series resistance was measured by a 10 mV disruption of the membrane voltage in voltage clamp.

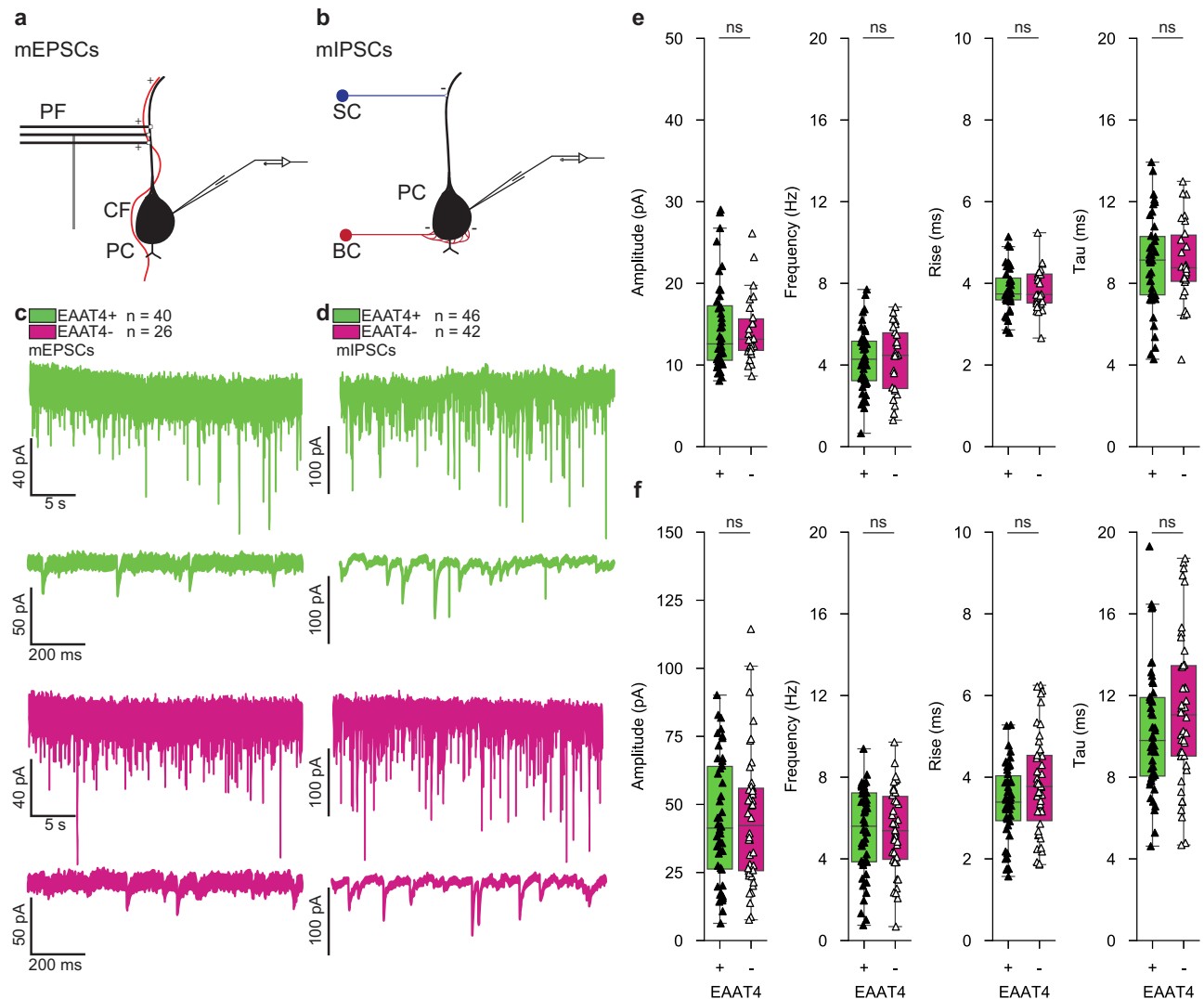

**Fig. 3 Miniature postsynaptic current kinetics and frequency do not differ between EAAT4+ and EAAT4- PCs. a, b** Schematic illustrations of the inputs to PCs that can induce miniature excitatory postsynaptic currents and miniature inhibitory postsynaptic currents. **c** 30-second example trace of a mEPSC recording and a 1-second example mEPSC trace of an EAAT4+ (green, top) and EAAT4- (red, bottom) PC. **d** Same as in **c**, but for mIPSCs. **e, f** No differences are observed between EAAT4+ and EAAT4- PCs in amplitude, frequency, rise time, nor tau (decay) for mEPSCs, (EAAT4+ $n = 40$, EAAT4- $n = 26$) (**e**) and mIPSCs (EAAT4+ $n = 46$, EAAT4- **n** = 42) (**f**). Black/white triangles represent median values of individual cells. Bars and boxes represent the 25th percentile. Differences between the amplitudes of mEPSCs and mIPSCs were determined with a Mann-Whitney U-test. All other data were assessed for differences with independent samples t-tests (Suppl. Table 3).

LTD recordings were performed in sagittal slices at physiological temperature (32-34 °C). PCs were selected from the entire cerebellar vermis based on their EAAT4 identity. PFs were stimulated in the molecular layer, and stimulation intensity adjusted to ~200–300 pA. CFs were stimulated in the granule cell layer, or the molecular layer, near the PC soma. Stimulation strength was adjusted to induce only a CF response without direct PC stimulation. Baseline EPSCs were recorded by a paired stimulus every 20 s for 10 minutes. LTD induction was performed in the current clamp, with 1 PF stimulation followed by a delayed CF stimulation after 120 ms. This was repeated every second for 5 minutes. Post-induction measurements were identical to baseline measurements, repeated for at least 30 minutes.

LTP recordings were performed in coronal slices at room temperature (21 °C ± 2), and in sagittal slices at physiological temperature (32–34 °C). Again, PCs were selected from the entire cerebellar vermis based on their EAAT4 identity. PFs were stimulated in the molecular layer, and stimulation intensity adjusted to ~200–300 pA. Baseline EPSCs were recorded by a paired stimulus (100 ms interval) every 20 s for 10 minutes (0.05 Hz). Subsequent LTP induction was done in current clamp, with 8 PF stimuli at 100 Hz every second for 5 minutes. Post-induction measurements were identical to baseline measurements, repeated for at least 30 min.

**Miniature postsynaptic currents**. Miniature postsynaptic currents were recorded in coronal slices at room temperature. PCs were selected from both the vermis and

hemispheres based on their EAAT4 identity. Once a PC of sufficient quality was patched, the cell was kept undisturbed in a voltage clamp for 3 minutes at −65 mV. This was done to allow the cesium internal solution to diffuse within the cell. After these initial 3 minutes, we applied slow capacitance and series resistance compensation of 40%. Subsequently, currents were recorded in voltage clamp for 120 seconds at −75 mV. When the recording was completed, the slow capacitance and series resistance compensation was removed, and the membrane potential was brought back up to -65 mV. The recording electrode was carefully removed from the cell, aiming to preserve the integrity of the cell membrane so that it could be immunohistochemically labeled and imaged. During mEPSC recordings, we added 1 $\mu$m of tetrodotoxin (TTX, tetrodotoxin citrate, Tocris, Abingdon, United Kingdom) and 100 $\mu$M of PTX to our ACSF. mIPSCs were recorded with 1 $\mu$m of TTX and 20 $\mu$M of NBQX (NBQX disodium salt, Tocris) added to our ACSF. PCs were excluded from analysis if their ratio between series and input resistance was >15%, or their holding current was < −500 pA. Additionally, PCs were excluded if the frequency of miniatures was extremely low (<0.5 Hz) or extremely high (>15 Hz). These cells were excluded because our automated analysis reported these very high or very low frequencies in noisy or unhealthy cells, respectively. In total, included 40 EAAT4+ /26 EAAT4- (9+ /11- excluded), and 46 EAAT4+ /42 EAAT4- (15+ /13- excluded), in the mEPSC and mIPSC datasets, respectively. The passive membrane properties of all PCs patched during miniature recordings were determined with a 10 mV disruption of the membrane voltage for 100 ms in the voltage clamp. The initial voltage was set at −65 mV. The resulting current responses were used to calculate the series (pipette), input, and

membrane resistances. Additionally, the membrane capacitance could both be estimated by Patchmaster software, and calculated from the previous experiment. This was done by calculating the area under the curve required to return to a steady state current after the initial 10 mV change. Measurements were repeated at least three times in each PC, and the results averaged.

**Immunohistochemistry**. All patched PCs were filled with biocytin present in our internal solutions. Upon completion of the relevant electrophysiological experiment, the patching pipette was carefully removed from the cell body. A slice, often with multiple cells patched, would then be removed from the ACSF bath, and placed into a jar of 4% paraformaldehyde (PFA). Slices were kept in 4% PFA in darkness for at least 24 hours. For immunohistochemistry, slices were removed from the 4% PFA solution, and washed 3 times for 10 minutes each in phosphate-buffered saline (PBS). Subsequently, slices were incubated in a blocking solution containing 10% normal horse serum (NHS) and 0.5% triton in PBS for 60 minutes. Finally, slices were incubated in a solution containing 1:400 streptavidin Cy3 (Jackson ImmunoResearch, Camebridgeshire, United Kingdom), 2% NHS, and 0.4% triton in PBS overnight at room temperature (±21 °C). On the second day, slices were washed 3 times in phosphate buffer (PB), DAPI for 10 minutes, and then a final 3 times in PB. Slices were mounted on cover slips. Note that during all procedures, slices were kept in darkness as much as possible so as to avoid bleaching of the eGFP.

For PLCB4 and aldolase C visualization, EAAT4-eGFP mice were perfused with 4% paraformaldehyde, and 50 μm thick coronal section were cut on a cryostat. Antibodies were diluted in PBS with 2% NHS and 0.4% triton. Antibodies used were aldolase C (1:1000; guinea pig, Synaptic Systems, Göttingen, Germany) and PLCβ4 (1:500, rabbit, Santa Cruz Biotechnology, CA). Slices were rinsed four times in PBS for 10 minutes. Subsequently, slices were blocked in a blocking solution containing 10% NHS and 0.5% triton for 60 minutes. Slices were incubated in the primary antibodies for 48–72 h at 4 °C. Slices were again rinsed four times in PBS. Secondary antibodies used were rabbit Cy3 (1:500, Jackson ImmunoResearch) and guinea-pig-Alexa-Fluor-647 (1:500, Jackson ImmunoResearch). Slices were incubated in secondary antibody with 2% NHS and 0.4% triton in PBS for at least 2 hours. Finally, slices were rinsed twice in PBS for 10 min, and subsequently mounted on cover slips.

**Imaging**. All fixed slices were imaged with an upright fluorescence microscope (Zeiss Axioscope, Zeiss, Oberkochen, Germany) and Zeiss software. In addition, some slices were also imaged using an upright confocal microscope (LSM 700, Zeiss). Labeled slices were used to confirm the EAAT4 identity of the patched PC if this identity was unclear in vitro, and to determine the exact location of the patched PC.

**Statistics and reproducibility**. Electrophysiological data was analyzed using custom Python scripts (Python 3.7+). Miniature postsynaptic currents were automatically analyzed using a deconvolution-based method[29]. Imaging data was manually reviewed (Zen Blue, Zen Black, ImageJ 1.53f51+). The location of patched cells was determined by comparing the locations of patched cells in our obtained images to a mouse brain atlas (Allen Brain Atlas, mouse.brain-map.org).

Statistical analysis was performed using custom Python scripts (Python 3.7+, SciPy 1.8.1+). All datasets were assessed for normality using Shapiro-Wilk tests. If data were normally distributed, a two-sided independent samples t-test was used to compare groups. If data were not normally distributed, a Mann-Whitney U test was used. Data variance was tested using Levene's test. Correlations between complex spike size and EPSC size were assessed with a Pearson's-r test. Results were considered significant if $p < 0.05$.

Statistical data were graphed using Python (matplotlib, seaborn). For boxplots, the box shows the quartiles of the dataset while the whiskers extend to show the rest of the distribution, except for points that are determined to be outliers using a method that is a function of the inter-quartile range. The central line between boxes represents the median. For all plots showing plasticity data over time, data from every cell were averaged per minute. Error bars represent the 95% confidence interval. No measures were taken to verify the reproducibility of the findings.

**Reporting summary**. Further information on research design is available in the Nature Portfolio Reporting Summary linked to this article.

## Data availability
Data will be made available upon reasonable request to the corresponding author.

## Code availability
Python scripts used in data analysis can be found on GitHub: https://github.com/s-voerman/EAAT4-Project

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

## Acknowledgements

The authors want to thank Elize Haasdijk for assistance with the immunohistochemistry. Financial support was provided by the H2020 European Research Council (ERC-Stg #680235) (MS); Nederlandse Organisatie voor Wetenschappelijk Onderzoek (OCENW.XS21.1.087), Erasmus MC Fellowship (JJW); Netherlands Organization for Scientific Research (NWO-ALW 824.02.001; CIDZ), the Dutch Organization for Medical Sciences (ZonMW 91120067; CIDZ), Medical Neuro-Delta (MD 01092019-31082023; CIDZ), INTENSE LSH-NWO (TTW/00798883; CIDZ, SV), ERC-adv (GA-294775; CIDZ) and ERC-POC (nrs. 737619 and 768914; CIDZ), NWO-Gravitation DBI2 (CIDZ), the Van Raamsdonk Foundation (CIDZ) and the NIN-Vriendenfonds for Albinism (CIDZ).

## Author contributions

Design of the experiments: S.V., B.H.A.U., M.S., J.J.W., C.I.D.Z. In vitro electrophysiology: S.V., B.H.A.U. Data analysis: S.V., B.H.A.U. Immunohistochemistry: S.V. Writing of the manuscript: C.I.D.Z., M.S., J.J.W., S.V.

## Competing interests

The authors declare no competing interests.
