## [Peer Review File · Communications Biology]

Reviewers' comments:

Reviewer #1 (Remarks to the Author):

Two population of Purkinje cells (zebrin-positive and -negative) differ in expression of many molecules and also forms distinct input (climbing fiber) and output connections. However, the functional significance of different molecular expressions of these Purkinje cells has not been fully clarified. Recently, several reports focused on this problem. This paper adds an important result on this issue timely. This study clarified the different long-term plasticity between zebrin-positive and -negative Purkinje cells in the mouse cerebellum. The experiments seem to have been done with expertise. Writing is excellent. The contents are easy to understand. Data are clearly shown in the paper. Statistics were generally clearly described unless mentioned below.

However, I have concerns about the novelty and impact of the reported findings.

1) Similar findings were originally reported by Wadiche and Jahr, 2004 (Nat Neurosci. 2005 8(10):1329-1334. doi: 10.1038/nn1539.). Their methods were not as sophisticated as in this study. They showed that Purkinje cells in lobule X, which are mostly zebrin-positive, barely show LTD, whereas Purkinje cells in lobule III, which are mostly zebrin-positive, shows LTD with the same protocol. Although they did not directly identify zebrin type of Purkinje cells, their essential results were almost the same as those shown in this paper.

2) This study is entirely descriptive. If any underlying mechanisms were explored, it would add some novelty and impact to this report.

3) Minor concerns.

Line 149, "local (DEC)": Meaning is not clear.

Line 223-224, "Additionally, PCs were excluded if the frequency of miniatures was extremely low (< 0.5 Hz) or extremely high (>15 Hz)." Why you can exclude them? Give the number of excluded cases (separately for EAAT4+ and EAAT4- PCs).

Line 244, "PB": Does this mean "PBS"?

Lines 249-256, "NHS": not defined,

Line 255, "A-647": What is it?

Lines 265-275, "Data and statistical analysis". In this section, the method of graphic presentation of statistic data should be clearly described. Are the box-whisker type presentation used in Figs 1,3 and their extended data figures according to the Turkey-type method? What is the length of the bars in Figs 1c and 2c,g.

"Data was assessed fortest. If the data was.... used" Then, the type of the test used in each analysis should be separately described in Figure legends.

Lines 269-270, "The location of patched cells was determined by comparing images to a mouse brain atlas (Allen Brain Atlas, mouse.brain-map.org)." I do not think it is possible. Do they have zebrin stripe images? Better describe what the authors did to confirm the location of the patched cells.

Legend of Figure 1

g, "Error bars represent 95% CI." What is "CI"?

h, "Error bars represent the 25th percentile": What is "the 25th percentile"? The box should also be defined.

h, "*** = $p < 0.001$, ** = $p < 0.01$." What test was used?

Reviewer #2 (Remarks to the Author):

The current, very succinct paper continues and extends the De Zeeuw laboratory's exploration and elucidation of the mechanisms underlying plasticity in the mouse cerebellum and the role of the cerebellar microzones in learning. As such, it adds an intriguing chapter to the role of LTD in Purkinje cells (PCs) located in so-called upbound and downbound microzones and the relevance of these zones, their cells, and the plasticity of these cells to behavioral phenomena including the vestibulo-ocular reflex and eyeblink conditioning. The major claim of the paper is that significant differences in the synaptic plasticity of upbound and downbound microzones are post synaptic

because a host of measures assessing presynaptic input did not show plasticity differences. The work is convincing and is certainly of interest to those investigating the relationship between cerebellar plasticity and learning if not the wider field.

There are some minor clarifications that could be included.

The paper suggests that the sequence of PF-CF stimulation with a gap of 120 ms used to induce LTD mimics classical conditioning. Although the pf-cf order is classical conditioning-like, the interstimulus interval is rather short for classical conditioning, and more importantly, classical conditioning would never happen if stimuli were presented once ever second for 5 minutes (300 "pairings". The correspondence should be down played.

It would be helpful to explain why LTP recordings were performed in coronal slices at room temperature ($21^{\circ}\text{C} \pm 2$), and in sagittal slices at physiological temperature ($32\text{-}34^{\circ}\text{C}$).

Reviewer #3 (Remarks to the Author):

This brief communication by Voerman et al. attempts to elucidate the underlying reasons for, essentially, why it is relatively "easier" to induce long-term depression in one Purkinje cell subpopulation compared to another. The brief results, which are worth publishing, are presented clearly and indicate that EPSC amplitude following a specific stimulation protocol was significantly more reduced in EAAT4-negative Purkinje cells.

General comments:

I) The conclusion on lines 74-75 is simply too broad. The presented results show that this one very specific stimulation protocol caused more of an EPSC amplitude reduction. How do the authors know that this means that this subpopulation has a "significantly enhanced capacity for LTD-induction"? The results only show this for the "1 PF + 1 CF after a 120 ms delay" protocol. This very manuscript makes reference to a study (16), which shows that different intervals are optimal in different regions and I am certain that the authors are aware of plenty of in vitro studies claiming different optima. In other words, how do the authors know that a smaller and possibly trivial difference or even the exact opposite results would not be obtained with a different protocol? Indeed, there may perchance not be strong reasons to assume such a difference, but are there strong arguments to simply assume that the results would hold?

In addition, how do the authors know that something other than the interval, e.g. the number of either PF or CF impulses, or intertrial interval, would give the same result (as the broad conclusion implies)? Certainly, the authors are aware of e.g. eyeblink conditioning studies, which shows that one CF pulse is even insufficient to induce a reduction in Purkinje cell simple spikes and instead even causes extinction (Rasmussen et al., *J Neurosci*, 2013;33(33):13436 –13440).

Unless the authors wish to do additional experiments where the protocol is varied in several ways, which I would understand is not tempting and may not be necessary, the conclusion in the manuscript should be made narrower and could be published with current data only. A lengthy discussion with references is probably not necessary in such a brief communication but a sentence or two should recognize this limitation in the concluding paragraph.

II) On lines 37-38 there is a statement of a PF-CF interval of 120 ms being compatible with "conditioning". It is unclear what is meant by "conditioning" and equally unclear that references 5 and 16 support it. For eyeblink conditioning, an interval of 120 ms seems "borderline" (e.g. Wetmore et al, *J Neurosci* 2014;34(5):1731–1737). I find clarification and slight re-phrasing rather important.

Minor comments:

a) The title is very broad and I would recommend a more narrow title that reflects the specific observation.

b) For the statement on line 22, certainly there are more appropriate references than reference number 5 which points to an unreviewed pre-print.

c) That the LTD experiments were performed in the presence of picrotoxin, not an inessential manipulation, should be mentioned in the main text.

Transparently signed review by:
Fredrik Johansson
Lund University

Point-by-point reply to the comments

Reviewer #1

Two population of Purkinje cells (zebrin-positive and -negative) differ in expression of many molecules and also forms distinct input (climbing fiber) and output connections. However, the functional significance of different molecular expressions of these Purkinje cells has not been fully clarified. Recently, several reports focused on this problem. This paper adds an important result on this issue timely. This study clarified the different long-term plasticity between zebrin-positive and -negative Purkinje cells in the mouse cerebellum. The experiments seem to have been done with expertise. Writing is excellent. The contents are easy to understand. Data are clearly shown in the paper. Statistics were generally clearly described unless mentioned below.

We thank the Reviewer for their compliments.

- However, I have concerns about the novelty and impact of the reported findings. Similar findings were originally reported by Wadiche and Jahr, 2004 (Nat Neurosci. 2005 8(10):1329-1334. doi: 10.1038/nn1539.). Their methods were not as sophisticated as in this study. They showed that Purkinje cells in lobule X, which are mostly zebrin-positive, barely show LTD, whereas Purkinje cells in lobule III, which are mostly zebrin-positive, shows LTD with the same protocol. Although they did not directly identify zebrin type of Purkinje cells, their essential results were almost the same as those shown in this paper.

Our work indeed follows in the footsteps of this seminal work. However, we also commit a considerable amount of effort to elucidate potential factors influencing the reported difference in postsynaptic LTD. First, we definitively demonstrate that the difference in LTD induction is specifically related to zebrin-identity rather than lobule location (as delineated previously). Second, while Wadiche and Jahr tested juvenile rats (P12-P21), we use adult mice (>P40), providing evidence against species-specificity and age-dependence. Third, we use a more physiologically relevant induction protocol using direct stimulation of climbing fibers instead of direct stimulation of the Purkinje cell as was done previously. Fourth, we report on both LTP and synaptic inputs in the form of miniature postsynaptic currents. Fifth, in the new manuscript, we now also present more data and analysis of the CF inputs (See new Extended Fig. 1E). From these additional control results we have concluded that we find a difference in postsynaptic PF-PC LTD, and that this difference is irrespective of inputs, but is likely a difference intrinsic to Z+ and Z- Purkinje cells. This final point relates to the still contentious issue in the cerebellar field of whether all Purkinje cells perform the same computation from different inputs. Our data imply that the fundamental computation of Purkinje cells is different based on their molecular identity. Indeed, as stated in the summarizing paragraph: 'We demonstrate that PCs do not all follow the same computation on differential inputs, but rather feature intrinsic differences that regulate differential plasticity outcomes to the same input.' Albeit there is always room for more mechanistic work, we hope that the Reviewer now finds our work sufficient to share it with the scientific field.

- This study is entirely descriptive. If any underlying mechanisms were explored, it would add some novelty and impact to this report.

Given this request, we have added additional analysis of the relationship between the size of the complex spike and the amount of LTD induced. We tested therefore the hypothesis that increased Ca²⁺ influx from a larger CS discharge influences the amount of LTD. However, we did not find a significant correlation between CS charge and LTD. The data from this analysis have now been added to Extended Fig. 1E, and we now highlight these data in the Results

section. Please note that we have adjusted the label on the y-axis of Extended Fig 1F from EPSC (%) to PPR, as was originally intended.

- Line 149, “local (DEC)”: Meaning is not clear.

DEC refers to the ethical committee in the Netherlands (Dier Experimentele Commissie or DEC). We apologize for the use of this unclear abbreviation. To resolve this, the ethics statement has been clarified by including more complete information about parties involved in ethical approval of the performed study.

- Line 223-224, “Additionally, PCs were excluded if the frequency of miniatures was extremely low (< 0.5 Hz) or extremely high (>15 Hz).” Why you can exclude them? Give the number of excluded cases (separately for EAAT4+ and EAAT4- PCs).

When cells were very noisy and unhealthy (with extremely high and/or low frequencies in traces), they were excluded. As a result, two cells were excluded from the mEPSC dataset. We now added the amount of excluded cells, with the amount of EAAT4+ and EAAT4- PCs separated. In addition, the criteria for exclusion have been added to the main text. Please also note that the amount of cells in each group reported in the legend of Figure 3 has now also been updated (after applying the exclusion criteria).

- Line 244, “PB”: Does this mean “PBS”?

We now provide the explanation for the abbreviation PB (phosphate buffer) in the text.

- Lines 249-256, “NHS”: not defined.

We now provide the explanation for the abbreviation NHS (normal horse serum) in the text.

- Line 255, “A-647”: What is it?

We now provide the explanation for the abbreviation AF-647 (Alexa-Fluor-647) in the text.

- Lines 265-275, “Data and statistical analysis”. In this section, the method of graphic presentation of statistic data should be clearly described. Are the box-whisker type presentation used in Figs 1,3 and their extended data figures according to the Tukey-type method? What is the length of the bars in Figs 1c and 2c,g?

In the boxplots the box shows the quartiles of the dataset, while the whiskers extend to show the rest of the distribution, except for points that are determined to be “outliers” using a method that is a function of the inter-quartile range. For all plots showing plasticity data over time, data from every cell were averaged per minute. Error bars represent the 95% confidence interval. This is now more clearly defined in the figure legends.

- “Data was assessed fortest. If the data was.... used” Then, the type of the test used in each analysis should be separately described in Figure legends.

The referenced sentence has been edited to improve its grammar and to clarify that the same statistical test for assessing normality was used for all data. In addition, the specific statistical tests used for each analysis is now separately described in each Figure legend. Finally, it is now more clearly stated in the figure legends what is meant by ‘CI’ and the 25th percentile of the boxes and bars.

- Lines 269-270, “The location of patched cells was determined by comparing images to a mouse brain atlas (Allen Brain Atlas, mouse.brain-map.org).” I do not think it is possible. Do

they have zebrin stripe images? Better describe what the authors did to confirm the location of the patched cells.

The Reviewer is right in that comparison to the atlas was largely redundant for the sagittal slices, as the lobular organization is very clear. In coronal slices, however, identification of the lobule in which a cell was located was more difficult, which made it useful to compare our data with images of the Allen brain atlas. We have tried to address this concern in the text by stating 'The lobular location of patched cells was verified by comparing the locations of patched cells in our obtained images to a mouse brain atlas (Allen Brain Atlas, mouse.brain-map.org).' We have now edited the text to clarify that this method allowed us to determine the location of PCs within the lobules. In addition, we have now made even more clear in the text that the zebrin identity was determined by the EAAT4 expression in our slices.

- Legend of Figure 1 g, "Error bars represent 95% CI." What is "CI"? h, "Error bars represent the 25th percentile": What is "the 25th percentile"? The box should also be defined. h, "**** = $p < 0.001$, ** = $p < 0.01$." : What test was used?

We have now added clear descriptions of CI, the box-plots, and used statistical tests to the legend.

Reviewer #2

- The current, very succinct paper continues and extends the De Zeeuw laboratory's exploration and elucidation of the mechanisms underlying plasticity in the mouse cerebellum and the role of the cerebellar microzones in learning. As such, it adds an intriguing chapter to the role of LTD in Purkinje cells (PCs) located in so-called upbound and downbound microzones and the relevance of these zones, their cells, and the plasticity of these cells to behavioral phenomena including the vestibulo-ocular reflex and eyeblink conditioning. The major claim of the paper is that significant differences in the synaptic plasticity of upbound and downbound microzones are post synaptic because a host of measures assessing presynaptic input did not show plasticity differences. The work is convincing and is certainly of interest to those investigating the relationship between cerebellar plasticity and learning if not the wider field.

We thank the Reviewer for their compliments.

- There are some minor clarifications that could be included. The paper suggests that the sequence of PF-CF stimulation with a gap of 120 ms used to induce LTD mimics classical conditioning. Although the pf-cf order is classical conditioning-like, the interstimulus interval is rather short for classical conditioning, and more importantly, classical conditioning would never happen if stimuli were presented once ever second for 5 minutes (300 "pairings". The correspondence should be down played.

We agree with the Reviewer that we should be careful about making a direct correlation between classical conditioning and our LTD results obtained *in vitro*; we are inducing LTD, not conditioning. However, we still believe our stimulus, and specifically, the timing of the complex spike is a relevant signal in this circuit. In classical conditioning, there is a marked increase in the number of complex spikes around 100 - 120 ms after the conditioned stimulus (e.g., Ohmae and Medina et al., 2015; Ten Brinke et al., 2015). It is believed that the complex spike is linked to the conditioned stimulus and further enhances plasticity. Given the comment of the Reviewer, we have now toned down our phrasings of the general relevance for conditioning; instead, in the new version we highlight the specific relevance of the used time interval for both the *in vitro* induction protocols and conditioning.

- It would be helpful to explain why LTP recordings were performed in coronal slices at room temperature ($21^{\circ}\text{C} \pm 2$), and in sagittal slices at physiological temperature ($32\text{-}34^{\circ}\text{C}$).

Our goal was to test for replicability of our findings in very different experimental conditions. This point has now been explained better in the main text.

Reviewer #3

This brief communication by Voerman et al. attempts to elucidate the underlying reasons for, essentially, why it is relatively "easier" to induce long-term depression in one Purkinje cell subpopulation compared to another. The brief results, which are worth publishing, are presented clearly and indicate that EPSC amplitude following a specific stimulation protocol was significantly more reduced in EAAT4-negative Purkinje cells.

We thank the Reviewer for their compliments.

- The conclusion on lines 74-75 is simply too broad. The presented results show that this one very specific stimulation protocol caused more of an EPSC amplitude reduction. How do the authors know that this means that this subpopulation has a "significantly enhanced capacity for LTD-induction"? The results only show this for the "1 PF + 1 CF after a 120 ms delay" protocol. This very manuscript makes reference to a study (16), which shows that different intervals are optimal in different regions and I am certain that the authors are aware of plenty of in vitro studies claiming different optima. In other words, how do the authors know that a smaller and possibly trivial difference or even the exact opposite results would not be obtained with a different protocol? Indeed, there may perchance not be strong reasons to assume such a difference, but are there strong arguments to simply assume that the results would hold?

The Reviewer is right in that there is indeed a wide variety of protocols used for LTD induction in the literature. However, it is important to note is that they all, to a greater or lesser extent, 'work' (they induce some negative change in synaptic strength). Whether our protocol is 'optimal' is not something that we are able to (or intend to) answer from our data. The intention was to compare plasticity in PCs with different molecular identities, but similar inputs. With our results, therefore, we think we can conclude that PCs with different identities respond differently to the same inputs (and that these differences are therefore probably intrinsic to PCs). Still, to incorporate this valid point of the Reviewer we have now refined the description of our conclusion. We now state: 'Our plasticity data highlight that individual PLCB4+ PCs, which are negative for EAAT4, differentially respond to similar inputs. Indeed, when LTD is induced by stimulating the CF input 120 ms after PF stimulation, the expression of LTD in EAAT4- PCs is enhanced compared to that in EAAT4+ PCs.'

- In addition, how do the authors know that something other than the interval, e.g. the number of either PF or CF impulses, or intertrial interval, would give the same result (as the broad conclusion implies)? Certainly, the authors are aware of e.g. eyeblink conditioning studies, which shows that one CF pulse is even insufficient to induce a reduction in Purkinje cell simple spikes and instead even causes extinction (Rasmussen et al., J Neurosci, 2013;33(33):13436 –13440). Unless the authors wish to do additional experiments where the protocol is varied in several ways, which I would understand is not tempting and may not be necessary, the conclusion in the manuscript should be made narrower and could be published with current data only. A lengthy discussion with references is probably not necessary in such a brief communication but a sentence or two should recognize this limitation in the concluding paragraph.

We thank the Reviewer for bringing up this important point. It is indeed not at all uncommon (especially in the recent ~5 years) to find studies about LTD induction protocols, especially *in*

vitro, that argue that multiple climbing fiber signals are required to induce LTD in PF-PC synapses. It was not our intention to induce the strongest LTD, but to compare plasticity between PC subpopulations. In essence, we attempted to determine to what extent PCs could respond differently in terms of plasticity induction to the same inputs. We now adapted our conclusion to be more focused on the obtained result (see also our response to the previous point), and we added a section in the final paragraph, describing and citing how using multiple CF/PF signals may influence LTD induction.

- On lines 37-38 there is a statement of a PF-CF interval of 120 ms being compatible with "conditioning". It is unclear what is meant by "conditioning" and equally unclear that references 5 and 16 support it. For eyeblink conditioning, an interval of 120 ms seems "borderline" (e.g. Wetmore et al, J Neurosci 2014:34(5):1731–1737). I find clarification and slight re-phrasing rather important.

We appreciate this comment and we have rephrased our wording accordingly. More specifically, we have now toned down our wording on the correspondence between our findings and conditioning, and instead highlight the role of the timing of the complex spike, and how this corresponds to the increase in complex spike activity observed in conditioning. We thus highlight the relevance of this time interval to both *in vitro* induction protocols as well as conditioning; this should render the references used more logical. In addition, we have added additional references to further reinforce this point (including Wetmore et al., 2014).

- The title is very broad and I would recommend a more narrow title that reflects the specific observation.

We appreciate this suggestion by the Reviewer, and we now narrowed it down to postsynaptic plasticity: "Postsynaptic plasticity of a Purkinje cell is determined by its molecular identity".

- For the statement on line 22, certainly there are more appropriate references than reference number 5 which points to an unreviewed pre-print.

This has been adjusted. An additional reference was added, and the peer-reviewed version of the mentioned paper was added.

- That the LTD experiments were performed in the presence of picrotoxin, not an inessential manipulation, should be mentioned in the main text.

We have added a statement that our extracellular solution contained picrotoxin.

REVIEWERS' COMMENTS:

Reviewer #1 (Remarks to the Author):

Two population of Purkinje cells (zebrin-positive and -negative) differ in expression of many molecules and also forms distinct input (climbing fiber) and output connections. However, the functional significance of different molecular expressions of these Purkinje cells has not been fully clarified. Recently, several reports focused on this problem. This paper adds an important result to this issue timely. This study clarified the different long-term plasticity between zebrin-positive and -negative Purkinje cells in the mouse cerebellum. The experiments seem to have been done with expertise. Writing is excellent. The contents are easy to understand. Data are clearly shown in the paper. Statistics were clearly described after some revisions made by the authors.

Although a similar study was performed before by Wadiche and Jahr (Nat Neurosci. 2005 8(10):1329-1334. doi: 10.1038/nn1539.), this study was performed in a different animal species (mice vs. rats) in a more systematic way with a direct comparison from zebrin-positive and -negative PCs, as mentioned in the rebuttal letter.

The authors have made enough revisions to improve the quality of the manuscript. I just noticed one minor concern:

mEPSCs were measured in 40 EAAT4+ PCs and 26 EAAT4- PCs (line 73). However, mEPSCs were NOT measured in $9+15=24$ EAAT4+ PCs and $11+13=24$ EAAT4- PCs (line 249). In line 249, the number of analyzed PCs should be written in addition to the number of excluded PCs for readers. Because the number of excluded PCs was large, $24/(24+40)$, $24/(24+26)$, whether it was a reliable experiment or not seems unclear to me.

Reviewer #2 (Remarks to the Author):

The changes and explanation in the "rebuttal" adequately address my concerns and comments.

Reviewer #3 (Remarks to the Author):

I thank the authors for their revision of this manuscript. All of my comments have been addressed prudently and I recommend publication in its current form.

Point-by-point reply to the comments

Reviewer #1

Two population of Purkinje cells (zebrin-positive and -negative) differ in expression of many molecules and also forms distinct input (climbing fiber) and output connections. However, the functional significance of different molecular expressions of these Purkinje cells has not been fully clarified. Recently, several reports focused on this problem. This paper adds an important result on this issue timely. This study clarified the different long-term plasticity between zebrin-positive and -negative Purkinje cells in the mouse cerebellum. The experiments seem to have been done with expertise. Writing is excellent. The contents are easy to understand. Data are clearly shown in the paper. Statistics were generally clearly described unless mentioned below.

We thank the Reviewer for their compliments.

- However, I have concerns about the novelty and impact of the reported findings. Similar findings were originally reported by Wadiche and Jahr, 2004 (Nat Neurosci. 2005 8(10):1329-1334. doi: 10.1038/nn1539.). Their methods were not as sophisticated as in this study. They showed that Purkinje cells in lobule X, which are mostly zebrin-positive, barely show LTD, whereas Purkinje cells in lobule III, which are mostly zebrin-positive, shows LTD with the same protocol. Although they did not directly identify zebrin type of Purkinje cells, their essential results were almost the same as those shown in this paper.

Our work indeed follows in the footsteps of this seminal work. However, we also commit a considerable amount of effort to elucidate potential factors influencing the reported difference in postsynaptic LTD. First, we definitively demonstrate that the difference in LTD induction is specifically related to zebrin-identity rather than lobule location (as delineated previously). Second, while Wadiche and Jahr tested juvenile rats (P12-P21), we use adult mice (>P40), providing evidence against species-specificity and age-dependence. Third, we use a more physiologically relevant induction protocol using direct stimulation of climbing fibers instead of direct stimulation of the Purkinje cell as was done previously. Fourth, we report on both LTP and synaptic inputs in the form of miniature postsynaptic currents. Fifth, in the new manuscript, we now also present more data and analysis of the CF inputs (See new Extended Fig. 1E). From these additional control results we have concluded that we find a difference in postsynaptic PF-PC LTD, and that this difference is irrespective of inputs, but is likely a difference intrinsic to Z+ and Z- Purkinje cells. This final point relates to the still contentious issue in the cerebellar field of whether all Purkinje cells perform the same computation from different inputs. Our data imply that the fundamental computation of Purkinje cells is different based on their molecular identity. Indeed, as stated in the summarizing paragraph: 'We demonstrate that PCs do not all follow the same computation on differential inputs, but rather feature intrinsic differences that regulate differential plasticity outcomes to the same input.' Albeit there is always room for more mechanistic work, we hope that the Reviewer now finds our work sufficient to share it with the scientific field.

- This study is entirely descriptive. If any underlying mechanisms were explored, it would add some novelty and impact to this report.

Given this request, we have added additional analysis of the relationship between the size of the complex spike and the amount of LTD induced. We tested therefore the hypothesis that increased Ca²⁺ influx from a larger CS discharge influences the amount of LTD. However, we did not find a significant correlation between CS charge and LTD. The data from this analysis have now been added to Extended Fig. 1E, and we now highlight these data in the Results

section. Please note that we have adjusted the label on the y-axis of Extended Fig 1F from EPSC (%) to PPR, as was originally intended.

- Line 149, “local (DEC)”: Meaning is not clear.

DEC refers to the ethical committee in the Netherlands (Dier Experimentele Commissie or DEC). We apologize for the use of this unclear abbreviation. To resolve this, the ethics statement has been clarified by including more complete information about parties involved in ethical approval of the performed study.

- Line 223-224, “Additionally, PCs were excluded if the frequency of miniatures was extremely low (< 0.5 Hz) or extremely high (>15 Hz).” Why you can exclude them? Give the number of excluded cases (separately for EAAT4+ and EAAT4- PCs).

When cells were very noisy and unhealthy (with extremely high and/or low frequencies in traces), they were excluded. As a result, two cells were excluded from the mEPSC dataset. We now added the amount of excluded cells, with the amount of EAAT4+ and EAAT4- PCs separated. In addition, the criteria for exclusion have been added to the main text. Please also note that the amount of cells in each group reported in the legend of Figure 3 has now also been updated (after applying the exclusion criteria).

- Line 244, “PB”: Does this mean “PBS”?

We now provide the explanation for the abbreviation PB (phosphate buffer) in the text.

- Lines 249-256, “NHS”: not defined.

We now provide the explanation for the abbreviation NHS (normal horse serum) in the text.

- Line 255, “A-647”: What is it?

We now provide the explanation for the abbreviation AF-647 (Alexa-Fluor-647) in the text.

- Lines 265-275, “Data and statistical analysis”. In this section, the method of graphic presentation of statistic data should be clearly described. Are the box-whisker type presentation used in Figs 1,3 and their extended data figures according to the Tukey-type method? What is the length of the bars in Figs 1c and 2c,g?

In the boxplots the box shows the quartiles of the dataset, while the whiskers extend to show the rest of the distribution, except for points that are determined to be “outliers” using a method that is a function of the inter-quartile range. For all plots showing plasticity data over time, data from every cell were averaged per minute. Error bars represent the 95% confidence interval. This is now more clearly defined in the figure legends.

- “Data was assessed fortest. If the data was.... used” Then, the type of the test used in each analysis should be separately described in Figure legends.

The referenced sentence has been edited to improve its grammar and to clarify that the same statistical test for assessing normality was used for all data. In addition, the specific statistical tests used for each analysis is now separately described in each Figure legend. Finally, it is now more clearly stated in the figure legends what is meant by ‘CI’ and the 25th percentile of the boxes and bars.

- Lines 269-270, “The location of patched cells was determined by comparing images to a mouse brain atlas (Allen Brain Atlas, mouse.brain-map.org).” I do not think it is possible. Do

they have zebrin stripe images? Better describe what the authors did to confirm the location of the patched cells.

The Reviewer is right in that comparison to the atlas was largely redundant for the sagittal slices, as the lobular organization is very clear. In coronal slices, however, identification of the lobule in which a cell was located was more difficult, which made it useful to compare our data with images of the Allen brain atlas. We have tried to address this concern in the text by stating 'The lobular location of patched cells was verified by comparing the locations of patched cells in our obtained images to a mouse brain atlas (Allen Brain Atlas, mouse.brain-map.org).' We have now edited the text to clarify that this method allowed us to determine the location of PCs within the lobules. In addition, we have now made even more clear in the text that the zebrin identity was determined by the EAAT4 expression in our slices.

- Legend of Figure 1 g, "Error bars represent 95% CI." What is "CI"? h, "Error bars represent the 25th percentile": What is "the 25th percentile"? The box should also be defined. h, "**** = $p < 0.001$, ** = $p < 0.01$." What test was used?

We have now added clear descriptions of CI, the box-plots, and used statistical tests to the legend.

- mEPSCs were measured in 40 EAAT4+ PCs and 26 EAAT4- PCs (line 73). However, mEPSCs were NOT measured in $9+15=24$ EAAT4+ PCs and $11+13=24$ EAAT4- PCs (line 249). In line 249, the number of analyzed PCs should be written in addition to the number of excluded PCs for readers. Because the number of excluded PCs was large, $24/(24+40)$, $24/(24+26)$, whether it was a reliable experiment or not seems unclear to me.

We have added the number of cells included to the methods section in line 249. It has also been made clearer that the number of excluded cells is $20/(40+26+20)$ (for mEPSC) and $28/(46+42+20)$, hopefully decreasing the doubts about the reliability of the experiments.

Reviewer #2

- The current, very succinct paper continues and extends the De Zeeuw laboratory's exploration and elucidation of the mechanisms underlying plasticity in the mouse cerebellum and the role of the cerebellar microzones in learning. As such, it adds an intriguing chapter to the role of LTD in Purkinje cells (PCs) located in so-called upbound and downbound microzones and the relevance of these zones, their cells, and the plasticity of these cells to behavioral phenomena including the vestibulo-ocular reflex and eyeblink conditioning. The major claim of the paper is that significant differences in the synaptic plasticity of upbound and downbound microzones are post synaptic because a host of measures assessing presynaptic input did not show plasticity differences. The work is convincing and is certainly of interest to those investigating the relationship between cerebellar plasticity and learning if not the wider field.

We thank the Reviewer for their compliments.

- There are some minor clarifications that could be included. The paper suggests that the sequence of PF-CF stimulation with a gap of 120 ms used to induce LTD mimics classical conditioning. Although the pf-cf order is classical conditioning-like, the interstimulus interval is rather short for classical conditioning, and more importantly, classical conditioning would never happen if stimuli were presented once ever second for 5 minutes (300 "pairings". The correspondence should be down played.

We agree with the Reviewer that we should be careful about making a direct correlation between classical conditioning and our LTD results obtained *in vitro*; we are inducing LTD, not

conditioning. However, we still believe our stimulus, and specifically, the timing of the complex spike is a relevant signal in this circuit. In classical conditioning, there is a marked increase in the number of complex spikes around 100 - 120 ms after the conditioned stimulus (e.g., Ohmae and Medina et al., 2015; Ten Brinke et al., 2015). It is believed that the complex spike is linked to the conditioned stimulus and further enhances plasticity. Given the comment of the Reviewer, we have now toned down our phrasings of the general relevance for conditioning; instead, in the new version we highlight the specific relevance of the used time interval for both the *in vitro* induction protocols and conditioning.

- It would be helpful to explain why LTP recordings were performed in coronal slices at room temperature ($21^{\circ}\text{C} \pm 2$), and in sagittal slices at physiological temperature ($32\text{-}34^{\circ}\text{C}$).

Our goal was to test for replicability of our findings in very different experimental conditions. This point has now been explained better in the main text.

Reviewer #3

This brief communication by Voerman et al. attempts to elucidate the underlying reasons for, essentially, why it is relatively "easier" to induce long-term depression in one Purkinje cell subpopulation compared to another. The brief results, which are worth publishing, are presented clearly and indicate that EPSC amplitude following a specific stimulation protocol was significantly more reduced in EAAT4-negative Purkinje cells.

We thank the Reviewer for their compliments.

- The conclusion on lines 74-75 is simply too broad. The presented results show that this one very specific stimulation protocol caused more of an EPSC amplitude reduction. How do the authors know that this means that this subpopulation has a "significantly enhanced capacity for LTD-induction"? The results only show this for the "1 PF + 1 CF after a 120 ms delay" protocol. This very manuscript makes reference to a study (16), which shows that different intervals are optimal in different regions and I am certain that the authors are aware of plenty of *in vitro* studies claiming different optima. In other words, how do the authors know that a smaller and possibly trivial difference or even the exact opposite results would not be obtained with a different protocol? Indeed, there may perchance not be strong reasons to assume such a difference, but are there strong arguments to simply assume that the results would hold?

The Reviewer is right in that there is indeed a wide variety of protocols used for LTD induction in the literature. However, it is important to note is that they all, to a greater or lesser extent, 'work' (they induce some negative change in synaptic strength). Whether our protocol is 'optimal' is not something that we are able to (or intend to) answer from our data. The intention was to compare plasticity in PCs with different molecular identities, but similar inputs. With our results, therefore, we think we can conclude that PCs with different identities respond differently to the same inputs (and that these differences are therefore probably intrinsic to PCs). Still, to incorporate this valid point of the Reviewer we have now refined the description of our conclusion. We now state: 'Our plasticity data highlight that individual PLCB4+ PCs, which are negative for EAAT4, differentially respond to similar inputs. Indeed, when LTD is induced by stimulating the CF input 120 ms after PF stimulation, the expression of LTD in EAAT4- PCs is enhanced compared to that in EAAT4+ PCs.'

- In addition, how do the authors know that something other than the interval, e.g. the number of either PF or CF impulses, or intertrial interval, would give the same result (as the broad conclusion implies)? Certainly, the authors are aware of e.g. eyeblink conditioning studies, which shows that one CF pulse is even insufficient to induce a reduction in Purkinje cell simple spikes and instead even causes extinction (Rasmussen et al., J Neurosci,

2013;33(33):13436–13440). Unless the authors wish to do additional experiments where the protocol is varied in several ways, which I would understand is not tempting and may not be necessary, the conclusion in the manuscript should be made narrower and could be published with current data only. A lengthy discussion with references is probably not necessary in such a brief communication but a sentence or two should recognize this limitation in the concluding paragraph.

We thank the Reviewer for bringing up this important point. It is indeed not at all uncommon (especially in the recent ~5 years) to find studies about LTD induction protocols, especially *in vitro*, that argue that multiple climbing fiber signals are required to induce LTD in PF-PC synapses. It was not our intention to induce the strongest LTD, but to compare plasticity between PC subpopulations. In essence, we attempted to determine to what extent PCs could respond differently in terms of plasticity induction to the same inputs. We now adapted our conclusion to be more focused on the obtained result (see also our response to the previous point), and we added a section in the final paragraph, describing and citing how using multiple CF/PF signals may influence LTD induction.

- On lines 37-38 there is a statement of a PF-CF interval of 120 ms being compatible with "conditioning". It is unclear what is meant by "conditioning" and equally unclear that references 5 and 16 support it. For eyeblink conditioning, an interval of 120 ms seems "borderline" (e.g. Wetmore et al, J Neurosci 2014;34(5):1731–1737). I find clarification and slight re-phrasing rather important.

We appreciate this comment and we have rephrased our wording accordingly. More specifically, we have now toned down our wording on the correspondence between our findings and conditioning, and instead highlight the role of the timing of the complex spike, and how this corresponds to the increase in complex spike activity observed in conditioning. We thus highlight the relevance of this time interval to both *in vitro* induction protocols as well as conditioning; this should render the references used more logical. In addition, we have added additional references to further reinforce this point (including Wetmore et al., 2014).

- The title is very broad and I would recommend a more narrow title that reflects the specific observation.

We appreciate this suggestion by the Reviewer, and we now narrowed it down to postsynaptic plasticity: "Postsynaptic plasticity of a Purkinje cell is determined by its molecular identity".

- For the statement on line 22, certainly there are more appropriate references than reference number 5 which points to an unreviewed pre-print.

This has been adjusted. An additional reference was added, and the peer-reviewed version of the mentioned paper was added.

- That the LTD experiments were performed in the presence of picrotoxin, not an inessential manipulation, should be mentioned in the main text.

We have added a statement that our extracellular solution contained picrotoxin.